# Strong correlational but no causal evidence on the link between the perception of scientific consensus and support for vaccination

**Gabriela Czarnek**\* , **Małgorzata Kossowska**

Institute of Psychology, Jagiellonian University, Krakow, Poland

\* gabriela.czarnek@uj.edu.pl

## Abstract

We examine the relationships between the perception of the scientific consensus regarding vaccines, and vaccine attitudes and intentions (N total = 2,362) in the context of COVID-19 disease. Based on the correlational evidence found (Study 1), perceived scientific consensus and vaccine attitudes are closely related. This association was stronger among people who trust (vs. distrust) scientists; however, political ideology did not moderate these effects. The experimental evidence (Studies 2–3) indicates that consensus messaging influences the perception of consensus; nonetheless, the effects on vaccine attitudes or intentions were non-significant. Furthermore, message aiming at reducing psychological reactance was similarly ineffective in changing attitudes as traditional consensus message.

## Introduction

Dr. Mike Ryan, Executive Director at the WHO Health Emergencies Programme, has recently said: *"What's shocked me most in this* [COVID-19] *pandemic has been that absence or loss of trust. (. . .) We need a social solution for future pandemics, much more than we need a technological solution"* [1]. The epidemic situation in many nations in late 2021, consistent with Dr. Ryan's observation, was the impetus for the current research. Specifically, even in countries where safe and effective COVID-19 vaccines were widely and freely available, sizeable groups of individuals remained unvaccinated [2]. At the time of running the current studies in December 2021, the average COVID-19 vaccination rate in high-income countries was 73%. Vaccine skepticism or hesitancy was especially strong in some countries: for example, while the vaccination rate was 67% in Switzerland, it was around 53% in Croatia and Poland, and merely 45% in Slovakia [2]. Crucially, the risk of COVID-19-related death in 2021 was around ten times higher among the unvaccinated compared to the vaccinated [3]. In this research, we investigated whether psychological interventions aimed at science communication can make a difference by focusing on people's perceptions of expert consensus.

Attitudes toward a wide range of issues are related to how much consensus people think there is among experts. This includes controversial topics such as climate change [4–7],

**Data Availability Statement:** The pre-registrations, data, and study materials are available at https://osf.io/dnjbu/.

**Funding:** The research was supported by the Copernicus Science Centre (Poland) https://www.kopernik.org.pl/en. The publication was supported by a grant from the Priority Research Area (Future Society: Behavior in Crisis Lab - Flagship Project) under the Strategic Programme Excellence Initiative at Jagiellonian University (Poland) https://phils.uj.edu.pl/en_GB/inicjatywa-doskonalosci. Founders had no role in study design, analysis, decision to publish, or preparation of the manuscript.

**Competing interests:** The authors have declared that no competing interests exist.

genetically-modified food [8], nuclear power [9], or medicinal cannabis [10]. Importantly, such a correlation has also been demonstrated in the context of infectious diseases [11, 12]. For example, early in the COVID-19 pandemic, people who believed that scientists agree that the disease is serious were more likely to support policies such as lockdowns or working from home for non-essential workers [12].

People, nevertheless, tend to underestimate the scientific consensus. According to polls, roughly half of the public believes that climate scientists are divided on whether climate change is occurring, or is caused by humans [13]. In contrast, several reports show that 97% of publishing climate change researchers agree that climate change is due to human activity [14]. Although climate skepticism is a fringe opinion among experts, these views are overrepresented in the media [15–17]. This, in turn, creates a perception of a false balance of opinion among the experts [18]. Similarly, presenting misleadingly 'balanced' views on the issue of vaccine safety leads to increased levels of uncertainty among the public [19, 20].

Happily, misperceptions regarding expert consensus can be corrected. Simply informing people that almost all scientists agree that climate change is human-caused increases the perception of scientific consensus [5, 7, 21]. Furthermore, according to the Gateway Belief Model [22], changes in consensus perception are associated with beliefs and attitudes more consistent with this consensus. Consensus messaging has also proven effective in the context of infectious diseases. Specifically, informing people that medical experts agree that the MMR vaccine is safe for children reduces worry, and belief that the vaccine causes autism; it also increases support for vaccination [11]. In the context of COVID-19, informing people of the fact that scientists agree that the disease is a serious one was linked to increased support for policies aimed at reducing virus transmission, such as mask mandates, quarantine for travelers, or bans on social gatherings [12].

Taken together, the abovementioned research indicates that people are likely to underestimate expert consensus and that consensus messaging should correct this, which, in turn, results in attitudes and intentions more consistent with the expert consensus. Thus, we asked in this research (1) whether the perception of scientific consensus is associated with attitudes toward vaccines and vaccination intentions; (2) whether short messaging changes the level of perception of the scientific consensus on vaccine safety and efficacy; (3) whether any increases in perceived consensus translate into more favorable vaccine attitudes and intentions; and (4) what the longevity of these effects, if any, might be.

The question arises, however, as to what factors may limit the effectiveness of consensus messaging. Although it has been proposed that communicating the consensus viewpoint neutralizes the politicization of certain topics [23], the evidence is quite mixed [24–26]. In the context of climate change, a recent meta-analysis shows that, while consensus messaging is effective among those both on the political Right and Left, it may cause greater psychological reactance among people on the political Right [27]. Still, given the limited reports on the role of ideology in consensus messaging about vaccines, and the fact that scientific issues are ideologically polarizing to varying degrees across contexts and times [28–31], it is not very clear what the role of ideology in the effectiveness of consensus messaging could be.

Furthermore, although trust in scientists tends to be correlated with ideology, the strength of this relationship also varies with time and between contexts [30, 32]. We propose that trust in scientists is key to the effectiveness of consensus messaging. Thus, we expect that consensus messaging is more effective in changing vaccine attitudes among those who trust scientists compared to those who distrust scientists. Our reasoning is that even if a person who distrusts scientists is informed that scientists agree on a particular issue and thus updates their perception of the level of consensus, such a person is still quite unlikely to concern themselves with what scientists have to say on the topic. In other words, consensus messaging might impact the

perception of levels of consensus but is unlikely to impact attitudes and behavior. This hypothesis is consistent with findings that show that consensus messaging is more effective among people with high deference to scientific authority as compared to low deference [33, 34] (although deference to scientific authority still does not always moderate the effectiveness of consensus messaging [8]).

In light of the literature on the asymmetry in consensus messaging receptivity, we posed further questions: (5) do consensus perception and messaging have similar effects on vaccine attitudes and intentions across the board, or do other factors, such as ideology or trust in scientists, moderate the effectiveness of consensus messaging?; (6) does acknowledging a plurality of opinion on vaccines, in conjunction with expert consensus messaging, improve the effectiveness of the intervention (as it might decrease reactance)?; and relatedly, (7) does consensus messaging result in psychological reactance, especially among certain groups?.

## Overview of the studies

We conducted one correlational study (Study 1) and two experiments (Studies 2 and 3). The summary of the research questions and studies with corresponding results is presented in Table 1. The studies were conducted with Polish samples that were matched to a national adult population on the basis of age, gender, and education (total N = 2,362, after exclusions). Studies 1 and 2 were run in December 2021 when COVID-19 vaccines were widely and freely available but at a time when nearly half (48%) of the population was still unvaccinated [2]. Concurrently, the country was experiencing peaks in COVID-19 cases and deaths [35]. Study 3 was run in June 2022, at which point the vaccination rate remained essentially unchanged (43% unvaccinated), but the epidemic situation had greatly improved [2, 35]. Nevertheless, given the uncertainty of future developments in the COVID-19 pandemic [36] and the further risk of novel infectious diseases in the future [37], we believe that it was still important to test the effectiveness of interventions aimed at increasing vaccine uptake and support.

Table 1. Summary of the research questions and findings.

| Research question | Study addressing the question | Findings |
|---|---|---|
| (1) Is the perception of scientific consensus associated with attitudes toward vaccines? | Study 1 | Yes: we found strong evidence for the association between perceived consensus and vaccine attitudes. |
| (2) Does short messaging change the level of perception of the scientific consensus on vaccine safety and efficacy? | Studies 2 and 3 | Yes: consensus messaging improves the perception of consensus. |
| (3) Does any increases in perceived consensus translate into more favorable vaccine attitudes and intentions? | Studies 2 and 3 | There is no evidence supporting the claim that increased perceived consensus translates into more favorable vaccine attitudes and intentions. |
| (4) What is the longevity of these effects, if any? | Study 2 | Partially yes: the consensus messaging effects on the perception of consensus lasts at least one week. In contrast, there is no evidence for the delayed effects of consensus messaging on vaccine attitudes. |
| (5) Do consensus perception and messaging have similar effects on vaccine attitudes and intentions across the board, or do other factors, such as ideology or trust in scientists, moderate the effectiveness of consensus messaging? | Studies 1 through 3 | Partially yes: Trust in scientists moderates the association between perceived consensus and vaccine attitudes; there is no evidence that ideology does.<br>Also, there is no evidence that trust, ideology, or priors moderate the effectiveness of consensus messaging. |
| (6) Does acknowledging a plurality of opinions on vaccines, in conjunction with expert consensus messaging, improve the effectiveness of the intervention (as it might decrease reactance)? | Study 2 | No evidence supports the claim that acknowledging the plurality of opinion improves consensus messaging effectiveness. |
| (7) Does consensus messaging result in psychological reactance, especially among certain groups? | Study 3 | Partially yes: consensus might increase psychological reactance, but there is no evidence that it is related to decreased effectiveness of consensus messaging. |

## Transparency and open science

For all the studies presented here, the hypotheses and analyses were pre-registered. The pre-registrations, data, and study materials are available at https://osf.io/dnjbu/. In both Study 1 (N = 58) and Study 2 (N = 78), we collected pilot data which was included in the final sample. These pilot tests were run to make sure the responses were properly saved and there were no technical issues. Collecting pilot data did not affect hypotheses but pilot testing to Study 1 showed that some participants might estimate scientific agreement as being at 0%; thus we decided to remove responses from such participants in the main study. The pre-registrations were submitted after the pilot data was collected, and this was disclosed in both pre-registrations. Study 3 was pre-registered before any data collection.

## Study 1

### Materials and methods

**Participants.**   The study was run between December 17th and 23rd, 2021. *A priori* power analysis suggested that to detect effects at 0.80 power in the most complex analysis (i.e. a mediated effect in which both effects for a and b paths are small with a bias-corrected bootstrap), the required sample is 462 [38]. However, we planned to collect a sample of 700 in order to match the sample size in previous studies on a similar topic where the per-country N was set to 700 [12]. We collected responses online through Pollster, a survey company, among their users. We requested quota sampling on age (above 18 years old), gender, and education to ensure a strong representation of adult Poles.

Seven hundred and six people took part in the study. In line with pre-registration (https://osf.io/35pn6), we removed seven participants whose initial perceived scientific consensus was as low as 0%. Re-analysis of the data with subjects reporting zero consensus and without controlling for demographic variables did not alter the results and conclusions. Details can be found in S1b, S3b, and S4b Tables in S1 File. It was unnecessary to remove any participants due to other pre-registered exclusion criteria (overly rapid response times, attention check failures, omissions of data on key variables) since all the participants met these criteria. The final sample was N = 699 (354 men, 344 women, 1 person identified as "other" with average age M = 47.91, SD = 16.38; and education M = 3.19, SD = 1.22); this figure was just shy of the target sample size of 700. We have removed responses from people identified as "other" from the analyses in Studies 1 and 2 due to statistical concerns. The specific reason for this was that estimated marginal means (which condition the predicted values of a DV on the nominal variables in the model) showed inflated confidence intervals due to such a small number of people identifying as "other". We fitted all the models with and without people identified as "other" and the results remained substantially unchanged. Nevertheless, we report models with those subjects removed.

**Materials.**   The main variables in this study were perceived scientific consensus, worry, belief, and policy support related to vaccines. Perceived scientific consensus was measured with one question ("To your knowledge (even if you don't know, please try to estimate): what % of medical scientists consider COVID-19 vaccines safe?"; 0%-100% on a sliding scale; M = 67.84, SD = 25.35). COVID-19 vaccine attitudes were measured as the degree of agreement with a series of statements (1 = "strongly disagree" to 5 = "strongly agree") and these responses formed three indices: (1) worry: two items measuring the perception that COVID-19 vaccines are harmful vs. safe, e.g., "I consider COVID-19 vaccines to be safe" (M = 2.84, SD = 1.20, Cronbach's α = 0.79); (2) belief: two items measuring the perceived efficacy of the vaccine, e.g., "I believe vaccines are an effective method of dealing with COVID-19" (M = 3.51,

**Table 2. Correlations between key variables in Study 1.**

| | Consensus perception | Worry | Belief | Policy support | Trust in scientists |
|---|---|---|---|---|---|
| Worry | -0.70 | | | | |
| Belief | 0.69 | -0.77 | | | |
| Policy support | 0.65 | -0.80 | 0.86 | | |
| Trust in scientists | 0.69 | -0.70 | 0.72 | 0.71 | |
| Ideology | -0.24 | 0.28 | -0.23 | -0.24 | -0.29 |

Note. All correlations were significant at p < .001.

SD = 1.19, Cronbach's α = 0.85); (3) policy support: five items measuring support for COVID-19 policies, e.g., "Entering cultural or sports events should be allowed only upon presentation of a COVID-19 vaccination certificate" (M = 3.22, SD = 1.42, Cronbach's α = 0.97).

We also measured political ideology and trust in scientists as potential moderators of the relationship between perceived scientific consensus and worry, belief, and policy support with regards to vaccines. Political ideology was measured with the Political Beliefs Questionnaire (PBQ; Cultural Beliefs subscale) [39]. The nine items measured the strength of agreement with policies such as "Christian values should be particularly protected in Poland" or "Poland should be primarily for Poles" (1 = "strongly disagree" to 5 = "strongly agree"; M = 2.61, SD = 0.99, Cronbach's α = 0.90). Trust in the scientific community was be measured with 3 questions: "How much do you trust that scientists: (1) work for the public good; (2) tell society the truth"; (3) "Overall, I trust scientists" (1 = "definitely not" to 5 = "definitely yes"; M = 3.42, SD = 1.09; Cronbach's α = 0.93). Correlations between the key variables are shown in Table 2.

**Procedure.** Participants gave their informed consent and then responded to a series of questions regarding trust in scientists and other sources of medical information, the perceived level of consensus regarding vaccination, attitudes toward vaccination, political ideology, and demographic variables. Apart from the measures just described, we also measured several additional variables, not related to the main research question. In Studies 1 and 2 these additional measures included perception of vaccines in general (not COVID-19 vaccine specifically), perceived social consensus regarding COVID-19 vaccines, COVID-19 threat, trust in institutions, political engagement, party preferences, need for cognitive closure (NFC). Further details are available in the pre-registrations. Finally, participants were debriefed and thanked; they also received points that could later be exchanged for a monetary reward.

**Analysis.** Prior to analysis, all continuous variables were rescaled between 0 and 1, and focal continuous predictors (perception of scientific consensus, ideology, trust in scientists) were additionally centered. Interactions were probed at +/-1 SD of the moderator. In all the analyses, we controlled for age, gender, and education, as pre-registered.

In all analyses presented in this paper, we used R [40] and RStudio [41] with lavaan [42], dplyr [43], tidyr [44], ggplot2 [45], haven [46], scales [47], emmeans [48], effects [49], multcomp [50], ggeffects [51], and sjPlot [52] and corrr [53] packages to clean, analyze, and present the data.

## Results

**Strong associations between perceived consensus and vaccine attitudes.** The perception of scientific consensus turned out to be strongly related to vaccine attitudes: negatively to vaccine worry (b = -0.79, SE = 0.03, t = -24.33, p < .001) and positively to the belief that the vaccine is effective (termed "belief" from here on; b = 0.76, SE = 0.03, t = 23.73, p < .001) and policy support (e.g., vaccine mandates; b = 0.84, SE = 0.04, t = 21.91, p < .001; all variables are

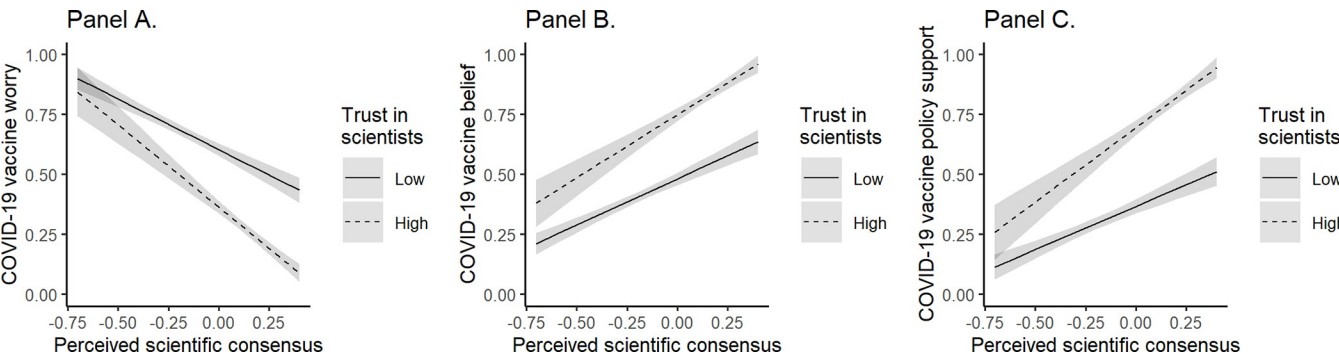

**Fig 1.** The effects of perceived consensus and trust in scientists on vaccine worry (Panel A), belief (Panel B), and policy support (Panel C). All variables have been scaled between 0 and 1; predictors (perceived scientific consensus and trust in scientists) were also centered. The slopes were probed at -1SD and +1SD of trust in scientists cores.

scaled between 0 and 1). The details are shown in S1a Table in S1 File. In addition, in line with the Gateway Belief Model [22], modeling relationships between the perception of consensus and vaccine attitudes using SEM showed that its effects were also indirect (S2 Table in S1 File).

**No evidence that ideology moderates association between perceived consensus and vaccine attitudes.** Furthermore, we found that right-wing ideology was positively related to worry (b = 0.13, SE = 0.03, t = 3.76, p < .001) and negatively to belief (b = -0.08, SE = 0.03, t = -2.28, p = .023) and policy support (b = -0.12, SE = 0.04, t = -2.90, p = .004) but that these effects were relatively small. More importantly, we found little support for the moderating role of ideology in the relationship between perceived consensus and vaccine attitudes. The effects of consensus on vaccine worry was somewhat less pronounced among those on the political Right (b = -0.69, SE = 0.04, 95%CI[-0.77, -0.61]) vs. Left (b = -0.86, SE = 0.05, 95%CI[-0.96, -0.76]) but these differences were, again, relatively small (b = 0.17, SE = 0.06, t(691) = 2.65, p = .008); the interaction between perceived consensus and ideology for other DVs was non-significant. The statistical details are presented in S3a Table and S1 Fig in S1 File.

**Trust in scientists moderates association between perceived consensus and vaccine attitudes.** As far as trusting scientists was concerned, this was negatively linked to worry (b = -0.44, SE = 0.04, t = -11.89, p < .001), but positively associated with belief (b = 0.50, SE = 0.04, t = 13.62, p < .001) and policy support (b = 0.61, SE = 0.04, t = 14.18, p < .001). As expected, we found that the associations between perceived consensus and the DVs (i.e., worry, belief, and policy support; Fig 1) were stronger for those who trust scientists as compared to those who do not (worry: b = -0.49, SE = 0.11, t = -4.65, p < .001; belief: b = 0.26, SE = 0.10, t = 2.56, p = .011; policy support: b = 0.48, SE = 0.12, t = 4.02, p < .001). The model details are presented in S4a Table in S1 File and the simple slopes analysis in section S1.3 in S1 File. More broadly, these findings align with the hypothesis that, while the association between ideology and science-related attitudes might vary between issues, cultures, or times [28, 30, 31, 54], trust in scientists seems closely linked to these attitudes.

## Study 2

In Study 1 we have established that the perception of scientific consensus is strongly associated with attitudes toward vaccines: lower worry that the vaccine is unsafe, higher beliefs that the vaccine is effective, and stronger support for vaccine policy. The goal of Study 2 was verifying whether short messaging changes the level of perception of the scientific consensus on vaccine safety and efficacy, and crucially, whether any increases in perceived consensus translate into

more favorable vaccine attitudes and intentions. We have also verified whether consensus perception and messaging have similar effects on vaccine attitudes and intentions across the board, or do other factors, such as ideology or trust in scientists, moderate the effectiveness of consensus messaging.

To estimate the effects of consensus messaging, we presented participants with consensus information regarding the safety of vaccines (consensus condition) or an unrelated medical consensus (control condition). Furthermore, we have evaluated whether intervention aiming at decreasing psychological reactance improves the effectiveness of consensus messaging. In order to do so, we added a third condition in which, besides informing participants about the scientific consensus around vaccine safety, it was acknowledged that there existed a plurality of opinions on this topic, i.e., that there is a great deal of information on the vaccines out there, and that public opinion is divided, but also that scientists agree that the vaccines are safe. Adding this condition was in response to evidence indicating that consensus messaging causes psychological reactance among some groups [26, 55]. By acknowledging that there might be diverging opinions on the topic in society, we aimed to decrease psychological reactance to consensus messaging, making such a communication strategy potentially more effective. This strategy is similar to restoration post-scripts [56], i.e., the message highlights that subjects are free to choose whether they will follow the scientific consensus advice or not.

The study's second goal was to assess the longevity of any consensus messaging effects. Specifically, we probed the perceived levels of scientific consensus and vaccine attitudes a week after the experimental manipulation.

## Materials and methods

### Participants

The study was run in two waves: the first took place between December 21$^{st}$ and 27$^{th}$, 2021, and the second after a week-long delay, i.e., between December 28$^{th}$, 2021 and January 4$^{th}$, 2022. Our intended sample size was N = 500. To account for data losses due to longitudinal design or inattention, we requested responses from 700 people, quota sampled on age, gender, and education from Pollster; 754 people took part in the study at wave 1 (control condition N = 237, consensus condition N = 265, consensus & pluralism condition N = 252) and 587 at wave 2 control condition N = 183, consensus condition N = 209, consensus & pluralism condition N = 195; which corresponds to 77%, 78%, and, and 77% of the original sample per condition, respectively).In line with the pre-registration (https://osf.io/myegr), we removed 134 participants whose rating of manipulation credibility was low (below 3 on a 5-point scale), 11 subjects whose initial perception of scientific consensus was as low as 0% (as it suggests expressive responding), and two participants with response times below the threshold (below 5 minutes in wave 1). Re-analysis of the data with subjects reporting zero consensus, low credibility, speeders, and without controlling for demographic variables did not alter the results and conclusions. Details can be found in S5b, S6b, S7b, and S8b Tables in S1 File). We did not remove any participants due to other pre-registered criteria since all of them met these (attention checks, no missing data on key variables). Thus, at wave 1, the final sample was N = 612 (321 men, 290 women, 1 person identified as "other"; with average age M = 47.44, SD = 16.83; and education M = 3.40, SD = 1.21). At wave 2, 587 participants participated in the study (i.e., 78% subjects accepted the invitation to partake in wave 2). However, after pre-registered exclusions (low credibility, low initial perception of scientific consensus, and overly rapid response times), 478 people were included in the analysis of responses at wave 2.

**Materials and procedure.** The design of Study 2 is depicted in Fig 2. We used the same DV measures as in Study 1; however, the design of the study was more complex. After giving

## Wave 1

> **Measurement of moderators**: ideology, trust in scientists
>
> **Pre-measurement** of all DVs
> - Perceived consensus
> - COVID-19 vax worry, belief, policy support

> Random assignment to **messaging condition**

| Control | Consensus | Consensus & pluralism |
|---|---|---|

> **Post-measurement** of all DVs
> - Perceived consensus
> - COVID-19 vax worry, belief, policy support

## Wave 2 (a week later)

> **Delayed measurement** of all DVs
> - Perceived consensus
> - COVID-19 vax worry, belief, policy support

**Fig 2. The design of Study 2.**

consent, responding to questions about political ideology (M = 2.55, SD = 1.02, Cronbach's α = 0.91), trust in scientists (M = 3.52, SD = 1.05, Cronbach's α = 0.92), and several other questions unrelated to the main research questions, participants responded to questions measuring perceived consensus (M = 73.81, SD = 21.51), worry (M = 2.51, SD = 1.11, Cronbach's α = 0.79), belief (M = 3.77, SD = 1.04, Cronbach's α = 0.83), and policy support (M = 3.56, SD = 1.32, Cronbach's α = 0.97) related to vaccines.

Next, we introduced an experimental manipulation of the scientific consensus. Subjects were randomly assigned to one of three conditions; in each condition, they were presented with information about the level of scientific consensus: (1) in the *consensus condition*, subjects were presented with the following question: "Did you know that 97% of medical scientists believe that COVID-19 vaccines are safe?", which is based on a manipulation used in previous research on scientific consensus [12, 57]; (2) in the *consensus & pluralism condition*, subjects were informed that "There is a lot of information on vaccination against COVID-19 and public opinion on this subject is divided. However, what we do know is that 97% of medical scientists believe COVID-19 vaccines are safe."; (3) in the *control condition*, subjects were shown this question: "Did you know that 97% of medical scientists believe that regular oral hygiene is essential for dental and gum health?". Nevertheless, our main analyses focus on comparing

*consensus* and *control* conditions; the *consensus & pluralism* condition was added to explore whether acknowledging controversy on the topic might help in disarming reactance and increase the effectiveness of consensus messaging.

After this manipulation, participants were presented with a few unrelated questions and responded again to questions related to the level of perceived scientific consensus regarding vaccination safety (M = 82.52, SD = 20.11); they were also asked about their worry (M = 2.53, SD = 1.11, Cronbach's α = 0.78), belief (M = 3.84, SD = 1.05, Cronbach's α = 0.82), and policy support (M = 3.55, SD = 1.34, Cronbach's α = 0.97) related to vaccines, as well as the perceived credibility of the manipulation; demographic information was also gathered.

A week later, participants were invited to partake in our survey at wave 2. They responded to questions about the perceived scientific consensus (M = 80.51, SD = 18.12), belief (M = 3.90, SD = 1.04, Cronbach's α = 0.79), worry (M = 2.49, SD = 1.11, Cronbach's α = 0.78), and policy support (M = 3.67, SD = 1.26, Cronbach's α = 0.96) related to vaccines. Finally, we provided debriefing to all the participants (including those who responded only in wave 1).

**Analysis.** In Study 2, we focused on the difference scores between pre- and post-manipulation responses (both measured at wave 1) as well as between pre-manipulation (wave 1) and delayed (wave 2) responses. We ran a series of regression analyses and SEM examining the relationship between experimental manipulations (dummy coded), and the differences scores in the perception of scientific consensus, vaccine worry, belief, and policy support. Furthermore, we investigated whether ideology and trust toward the scientific community moderate the effects of consensus messaging; the slopes were probed at +/- 1 SD from a mean. Continuous predictors were rescaled between 0 and 1, and the moderators were additionally centered. As pre-registered, in all analyses we control for age, gender, and education.

## Results

**Consensus messaging increases the perception of scientific consensus.** We found that the brief consensus messaging lifted participants' perception of scientific consensus in wave 1, and that these effects persisted a week later in wave 2. As shown in Fig 3, when compared to the control condition, participants' change scores in the consensus (wave 1: b = -9.64, SE = 1.46, t(605) = -6.62, p < .001; wave 2: b = -6.80, SE = 1.67, t(471) = -4.06, p < .001), and in the consensus & pluralism condition (b = -9.00, SE = 1.44, t(605) = -6.24, p < .001; b = -5.57, SE = 1.66, t(471) = -3.35, p = .003) were significantly higher; but the difference between consensus vs. consensus & pluralism was non-significant (b = 0.64, SE = 1.45, t(605) = 0.44, p = .898; b = 1.23, SE = 1.67, t(471) = 0.736, p = .742); the model details are presented in S5a Table in S1 File. These results show that, on average, the perception of scientific consensus increased by around 9 percentage points right after messaging, and around 6 percentage points a week later, as compared to the measurement before consensus messaging occurred. An exploratory analysis showed that the difference in these effects between the two waves was significant (b = -4.23, SE = 0.74, t(346.01) = -5.71, p < .001).

Next, we examined whether political ideology and trust in scientists moderate the effects of consensus messaging on the perceived consensus. For both waves, the effects of ideology on change in perceived consensus and the interaction with consensus messaging were non-significant; the model details are presented in S5a Table in S1 File. In contrast, trust in scientists moderated the effects of consensus messaging on perceived consensus in both waves. As shown in Fig 4, people with low vs. high trust in scientists updated their perception of scientific consensus more in the consensus condition (wave 1: b = 9.72, SE = 2.06, t(602) = 4.73, p < .001; wave 2: b = 8.32, SE = 2.39, t(468) = 3.48, p < .001) and in the consensus & pluralism condition (wave 1: b = 7.66, SE = 2.00, t(602) = 3.83, p < .001; wave 2: b = 8.02, SE = 2.37, t(468) =

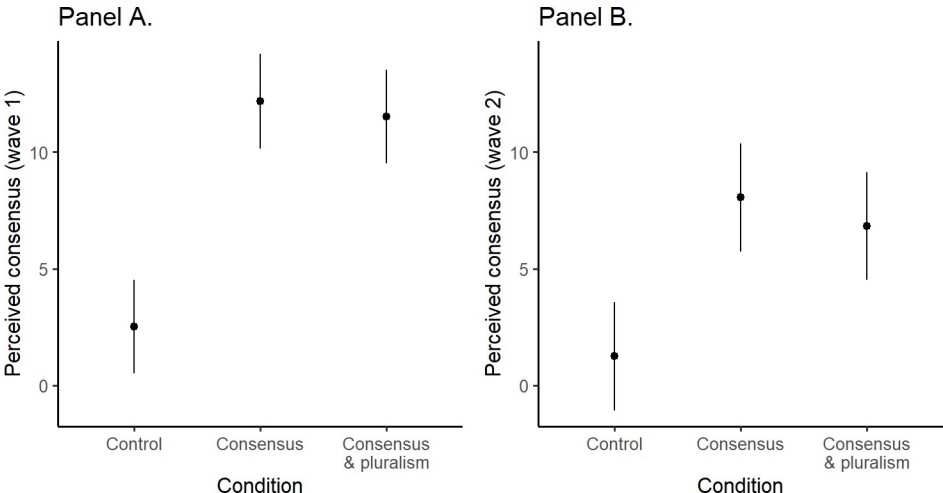

**Fig 3.** The effects of consensus messaging on perceived consensus in Study 2 measured at wave 1 (Panel A) and at wave 2 (Panel B). The y-axis represents change scores.

3.38, p < .001), but the difference in the control condition was non-significant (wave 1: b = 1.45, SE = 2.01, t(602) = 0.72, p = .470; wave 2: b = 0.37, SE = 2.33, t(468) = 0.16, p = .874; for further details see S5A Table in S1 File). These effects are due to low prior perceptions of consensus among people with low trust in scientists.

**No evidence for changes in attitudes toward the vaccine after consensus messaging.** We next focused on the effects of the consensus messaging on vaccine attitudes, i.e., worry, belief, and policy support with regards to vaccines. These effects turned out to be non-significant for ten out of the 12 effects analyzed. Specifically, the effect of consensus messaging vs. control condition was non-significant for belief (b = 0.01, SE = 0.07, p = 0.896 at wave 2), worry (b = 0.05, SE = 0.06, p = 0.408 at wave 1; b = 0.05, SE = 0.08, p = 0.551 at wave 2) and policy support (b = -0.01, SE = 0.04, p = 0.848 at wave 1; b = -0.05, SE = 0.06, p = 0.443 at wave 2, Similarly, the effect of consensus and pluralism messaging vs. control condition was non-

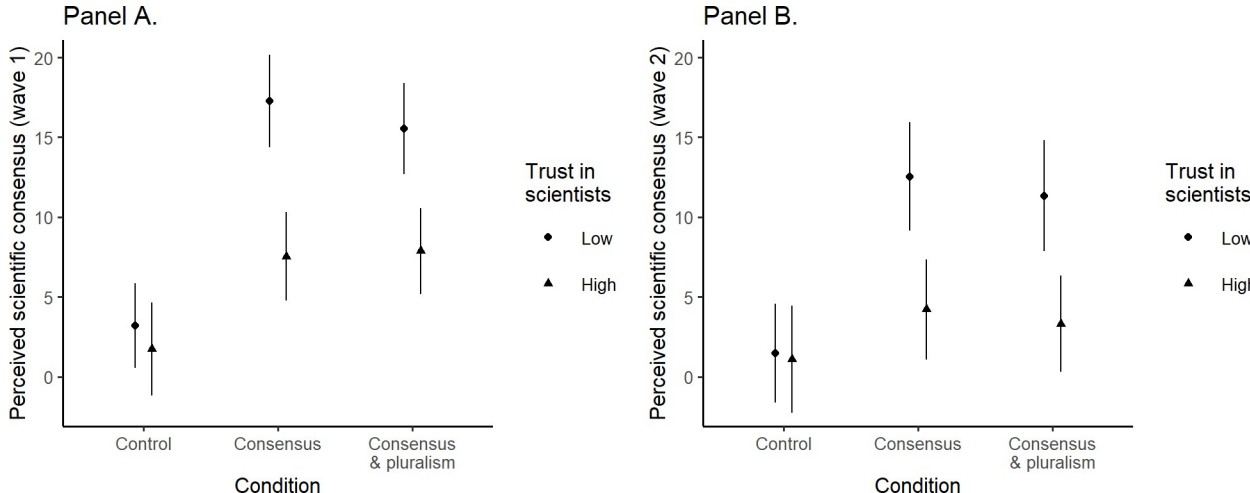

**Fig 4.** The effects of consensus messaging and trust in scientists on perceived consensus in wave 1 (Panel A) and wave 2 (Panel B) in Study 2. The y-axis represents change scores.

significant for belief (b = 0.03, SE = 0.05, p = 0.525 at wave 1; b = 0.12, SE = 0.07, p = 0.086 at wave 2), worry (b = 0.02, SE = 0.05, p = 0.703 at wave 1), and policy support (b = 0.06, SE = 0.04, p = 0.100 at wave 1; b = -0.03, SE = 0.06, p = 0.601 at wave 2). For further model details see S6A Table in S1 File).

The two remaining significant effects of consensus messaging were found for belief in wave 1 (consensus vs. control condition: b = 0.10, SE = 0.05, t(605) = -2.07, p = 0.039) and worry in wave 2 (consensus & pluralism vs. control condition: b = 0.21, SE = 0.08, t(471) = 2.72, p = .007). However, these effects were inconsistent with the theoretical model and not systematic across different consensus conditions. Further details are presented in section S2.1.2 of the S1 File and S2 Fig (S1 File). We also did not find any evidence that the effectiveness of consensus messaging on vaccines varies between people espousing different ideologies or with different levels of trust in scientists. For more on these analyses, see S7a (ideology) and S8a (trust) Tables in S1 File.

Despite the abovementioned results, it might be the case that the effects of consensus messaging are indirect. According to the Gateway Belief Model, a consensus message impacts attitudes through a change in perceived consensus. For that reason, we fit SEM (ML estimator, 5,000 iterations) with the paths from experimental manipulation on perceived consensus, and perceived consensus through belief and worry on policy support (specified in a way similar to Kerr & Van der Linden, 2021, Study 2). Due to the presence of three conditions, we fit three SEM models comparing each contrast, i.e., (i) consensus vs. control conditions, (ii) consensus & pluralism vs. control conditions, and (iii) consensus vs. consensus & pluralism conditions. Our analyses showed that a model fit was not satisfactory for half of the models (see S9 Table in S1 File). Nevertheless, even for the models with satisfactory fit, there was no evidence for the indirect effects of consensus messaging on vaccine attitudes; see S10-S15 Tables in S1 File for more details on this.

## Study 3

In Study 2 we found that consensus messaging increases the perception of scientific consensus regarding vaccine safety, but there was no evidence that this, in turn, translates into more positive attitudes toward vaccines. That was the case for the standard consensus message as well as the consensus & pluralism condition. We would like to make several observations with regards to Study 2.

First, we used a repeated-measure design in which we probed participants' attitudes on three occasions (before and after messaging, and a week later). Although a repeated-measure design is essential according to the Gateway Belief Model [22], others have suggested that such a design might bias people's responses. For example, Landrum and colleagues (2021) used a repeated-measure design and found the effects of consensus messaging on perceived consensus, but not on the attitudes of interest. They argued that exposing participants to the DVs before consensus messaging undermines message efficacy, which they call *sensitization* effects. Relatedly, some researchers suggest that consensus messaging might cause psychological reactance [26, 58]. Unfortunately, we did not measure psychological reactance in Study 2; in any case, there was no evidence that messaging which acknowledged a plurality of opinion on the vaccines was superior to standard consensus messaging. Secondly, in Study 2, we placed our focus on vaccine safety; however, an important dimension of vaccine attitudes might also be their efficacy, e.g., [59]. Furthermore, although our sample was relatively large, the power analysis accounted for the comparison between two groups, whereas we analyzed responses emerging from three conditions. Another caveat is that, pre-registered, we excluded people who perceived the consensus messaging as low on credibility (18% of the original sample). As it

transpired, consensus messaging did not improve vaccine attitudes, even among people who found the messaging at least somewhat credible. On the other hand, testing its effectiveness among entire sample including people who perceive messaging as low on credibility, might be a more appropriate way to analyze the data. We will address these shortcomings in Study 3.

Specifically, to verify the efficacy of consensus messaging and test for a sensitization effect, we introduced two experimental manipulations: consensus messaging (control vs. vaccine consensus) and study design ("pre-post" vs."post-only"). We used consensus messaging similar to that in Study 2 but added information about vaccine efficacy (for details, see the Method section). In addition, we decided to drop the consensus & pluralism condition in order to reduce the number of conditions, and owing to a lack of any systematic differences between the consensus vs. the consensus & pluralism conditions being observed in Study 2. To implement the manipulation of the study design, we introduced a "pre-post" condition in which subjects responded to the DVs before and after manipulation, i.e., as in Study 2, and a "post-only" condition in which participants responded to the main DVs only after the consensus messaging manipulation. We also added measures of psychological reactance and gauged vaccination intentions (in terms of either the first dose of the vaccine among the unvaccinated, or a booster shot among the vaccinated). Finally, we added the analysis of moderation by priors, on top of the analysis of the moderation by ideology and trust in scientists.

## Materials and methods

### Participants

The study was run between June 23$^{rd}$ and 28$^{th}$, 2022. As already mentioned, the sample required for the most complex analysis is 462. Thus, our intended sample size was N total = 1,000 as we planned to analyze data separately for the pre-post and post-only condition; we also accounted for potential data losses. We requested responses from 1,000 users, quota sampled on age, gender, and education, from Panel Ariadna (an online survey company) and 1,073 people took part in the study.

In line with the pre-registration (https://osf.io/hwm6k), we removed nine participants with overly rapid response times (less than 5 minutes in "pre-post" design conditions, and 3.33 in the "post-only" condition), and 16 people whose initial perceived scientific consensus was as low as 0%. Re-analysis of the data with subjects reporting zero consensus, speeders, and without controlling for demographic variables did not alter the results and conclusions. Details can be found in S16b, S17b, S18b, S19b, S20b, S21b, S22b, S23b, S24b, S25b, S31b, and S32b Tables in S1 File. We did not remove any participants due to other pre-registered exclusion criteria as all subjects met them (attention checks, no missing data on key variables). The final sample was N = 1,051 (484 men, 567 women; with average age M = 45.48, SD = 15.52; and education M = 3.30, SD = 1.22). Our final sample is large enough to detect even the most complex hypothesized relationships.

**Materials and procedure.** The design of Study 3 is depicted in Fig 5. After giving consent, responding to questions about political ideology (M = 2.80, SD = 0.88, Cronbach's α = .88) and trust in scientists (M = 3.12, SD = 1.08, Cronbach's α = .92), participants were briefly reminded that, although most of the COVID-19 restrictions have been lifted, we are not over the pandemic yet (this was to increase the salience of the topic given that most of the public's attention at the time focused on the war in Ukraine). After this, participants were randomly assigned to a "pre-post" (N = 508) or a "post-only" design condition (N = 543). Then, only participants in the "pre-post" design condition responded to the key DVs, i.e., questions measuring perceived level of scientific consensus (M = 66.52, SD = 24.22), worry (M = 2.84, SD = 1.3 Cronbach's α = 0.88), belief (M = 3.36, SD = 1.21 Cronbach's α = 0.84), and policy support

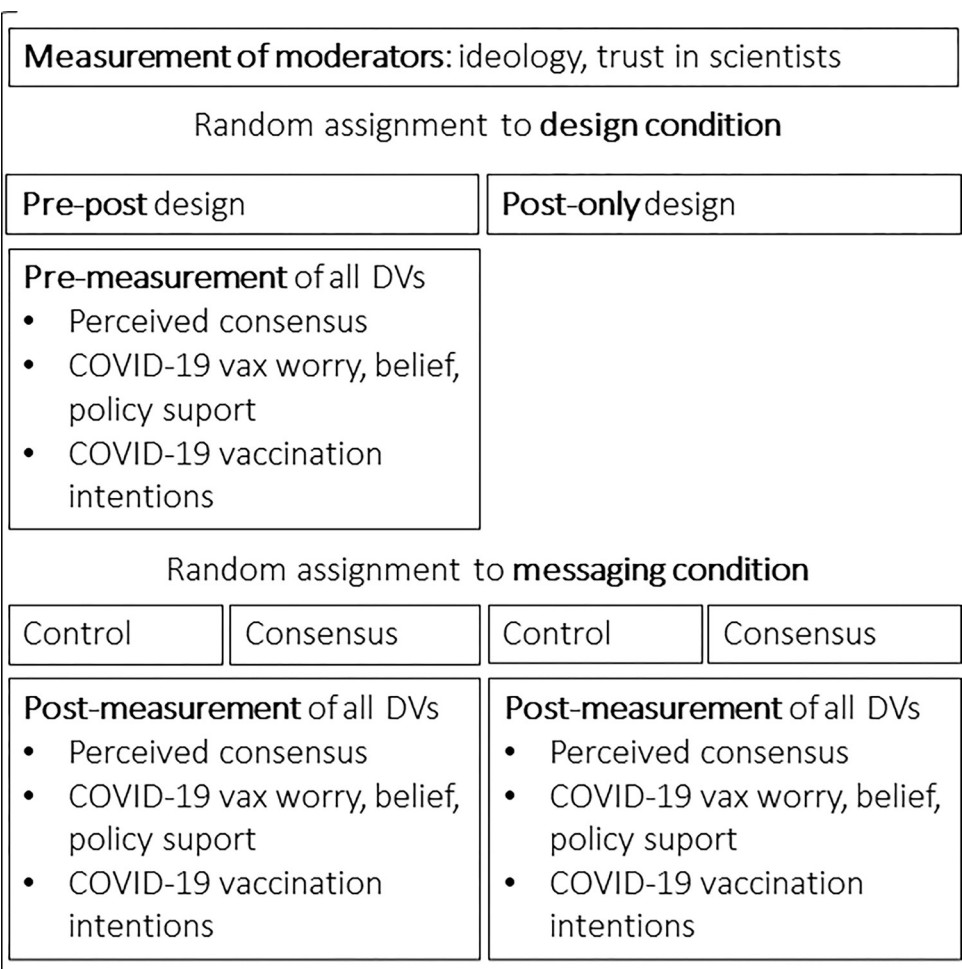

**Fig 5. The design of Study 3.**

(M = 2.92, SD = 1.4 Cronbach's α = 0.97) related to vaccines. The questions regarding worry, belief, and policy support were exactly as in Studies 1 and 2 but we changed the wording of the question slightly for the perceived scientific consensus question: "To your knowledge (even if you don't know, please try to estimate): What % of medical scientists considers vaccines to be safe and effective in protecting against severe symptoms of the disease?". We additionally measured the perceived scientific certainty regarding vaccines ("Do you think science provides certainty on whether COVID-19 vaccination is safe and effective?", 1 = "definitely not" to 5 = "definitely yes"; M = 3.09, SD = 1.15). Participants were also asked whether they had already received any dose of the vaccine (375 people in the "pre-post" design condition were vaccinated and 168 were not) and about their intention to receive a first dose of the vaccine (among the unvaccinated; 1 = "Definitely not" to 5 = "Definitely yes"; M = 1.55, SD = 0.80) or a booster shot if that was recommended by experts (among the vaccinated; 1 = "Definitely not" to 5 = "Definitely yes"; M = 3.66, SD = 1.35).

Next, we introduced an experimental manipulation with regards to the scientific consensus. Subjects were randomly assigned to the control or the consensus messaging condition both in the "pre-post" design condition (control messaging N = 249, consensus messaging N = 259) and in the "post-only" design condition (control messaging N = 271, consensus messaging N = 272). Information presented in the control condition was exactly as in Study 2. The

manipulation in the consensus condition was: "Did you know that 97% of medical scientists agree that COVID-19 vaccines are safe and effective in preventing the risk of severe disease symptoms and hospitalization?"

After manipulation, participants were given a few questions unrelated to the main goal of the study and responded to the key DVs, i.e., perceived scientific consensus regarding vaccines safety and effectiveness (M = 73.12, SD = 24.58), vaccine worry (M = 2.83, SD = 1.24 Cronbach's α = 0.84), belief (M = 3.34, SD = 1.19 Cronbach's α = 0.84), and policy support (M = 2.90, SD = 1.41 Cronbach's α = 0.97), as well as behavioral intentions to get vaccinated (M = 1.60, SD = 0.85) or boosted (M = 3.64, SD = 1.33); we also asked about the perceived scientific certainty regarding the vaccines (M = 3.16, SD = 1.17). Next, we measured the perceived credibility of the manipulation (control messaging: M = 3.83, SD = 0.85; consensus messaging: M = 3.29, SD = 1.14) and psychological reactance to the messaging: emotional reactance ("To what extent did the information on the scientific consensus regarding oral hygiene/COVID-19 vaccines annoy you?"), cognitive reactance (". . .make you feel that others are trying to impose their opinions on you?"), and counterarguing (". . . make you think about why you disagree with the presented information?"); the questions were designed to tap different dimensions of psychological reactance (based on Moyer-Gusé and Nabi [60]; however, exploratory factor analysis showed that they were very closely related and loaded on just one dimension; thus, we analyzed the average scores of the three items (M = 2.40, SD = 0.93 Cronbach's α = 0.85). We also asked several other questions unrelated to the main question of the study:: perceived invulnerability to COVID-19 disease, threshold of scientific consensus necessary to trust the COVID-19 vaccine, perceived social consensus regarding COVID-19 vaccines, political engagement, party preferences, economic beliefs, NFC, and perceived threat related to war in Ukraine. Further details are available in the pre-registrations. Finally, they were debriefed and thanked. Participants received activity points that could then be exchanged for rewards of their choice.

**Analysis.**   The analysis was very similar to the one in Study 2, except that in the "post-only" design condition, we did not analyze change scores, but raw responses given after consensus messaging manipulation. We ran most analyses separately for "pre-post" and "post-only" design conditions, except for the analyses of sensitization effects and psychological reactance. As previously, continuous predictors were rescaled between 0 and 1, and the moderators were additionally centered. The slopes were probed at +/- 1 SD from a mean. As pre-registered, in all analyses, we controlled for age, gender, and education.

## Results

**No evidence of sensitization effects.**   To detect any sensitization effects, we ran regression analyses using responses from the post-measurement of attitudes and intentions. We included the main and interactive effects of messaging (control vs. consensus) and design (pre-post vs. post-only) for each model. Statistically significant effects of the design and design x messaging interaction would suggest sensitization effects [10, 61]; especially problematic would be statistically significant interaction term as it would suggest that the effects of consensus messaging manipulation depends on whether participants responded to DV questions beforehand. However, no statistically significant effect of the design was found, nor of the interaction between study design and consensus messaging for any of the DVs (i.e., perceived consensus, vaccine beliefs, worry, policy support, and vaccination intentions). In other words, we did not find any evidence of sensitization effects; more details are presented in S16a Table in S1 File.

**Replicating findings that consensus messaging increases the perception of scientific consensus.**   Next, we ran models, similar to those in Study 2, separately for the "pre-post" and

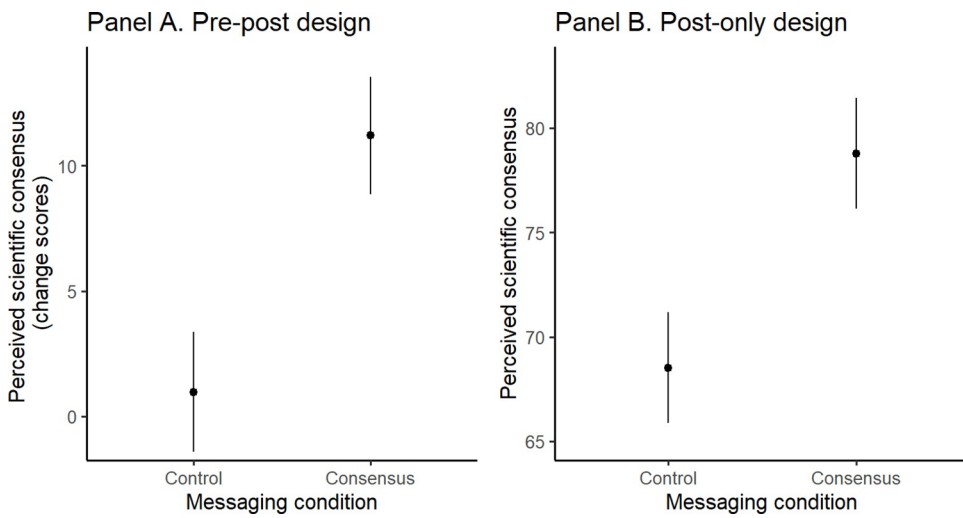

**Fig 6.** The effects of consensus messaging on perceived consensus in Study 3 in pre-post design (Panel A) and post-only design conditions (Panel B).

"post-only" design conditions. Here, we again found that consensus messaging lifted the participants' perception of scientific consensus in both the "pre-post" and "post-only" design conditions; the model details are shown in S17 Table in S1 File. On average, the perception of scientific consensus changed by around 10 percentage points after consensus messaging, as shown in Fig 6. This extends the findings from Study 2 by showing that consensus messaging is effective in increasing the perception of scientific consensus irrespective of our decision to include or exclude people who perceived the message as not credible.

The findings on the role of ideology, trust in scientists, and priors in the "pre-post" design condition replicate and extend the findings from Study 2. In the "pre-post" design condition, the moderating effects of ideology were non-significant (b = -4.80, SE = 7.80, t(501) = -0.62, p = .539), while trust in scientists (b = -19.32, SE = 6.32, t(501) = -3.06, p = .002) and priors (b = -14.94, SE = 6.54, t(501) = -2.28, p = .023) were significant. Specifically, people with low vs. high trust in scientists updated their perceptions of consensus more in response to consensus messaging (b = 8.82, SE = 2.35, t(501) = 3.75, p < .001); but not in the control condition (b = -1.42, SE = 2.46, t(501) = -0.58, p = .564; Fig 7, Panel A). We observed a similar pattern of results for people with low vs. high priors (details are in section S3.2 of the S1 File and S3 Fig (S1 File)). In the "post-only" design condition, both ideology (b = -27.92, SE = 6.11, t(536) = -4.57, p < .001) and trust in scientists (b = 47.36, SE = 4.57, t(536) = 10.36, p < .001) were associated with perceived consensus; the latter also interacted with consensus messaging (b = -17.76, SE = 6.17, t(536) = -2.88, p = .004). As shown in Fig 7, Panel B, for people with high trust in scientists, consensus messaging only slightly increased perceived consensus (b = -6.60, SE = 2.37, t(536) = 2.78, p = .006) but for those low on trust in scientists, consensus messaging made a more substantial difference (b = -16.40, SE = 2.42, t(536) = -6.77, p < .001). Further model details are presented in S17a Table in S1 File.

**Replicating null effects of consensus messaging on vaccine attitudes and intentions.** Notwithstanding the effects detected on perceived consensus, the effects of consensus messaging on vaccine attitudes and vaccination intentions were non-significant both in the "pre-post" design (change scores) and in the "post-only" design conditions. Specifically, in the "pre-post" design condition the effects of consensus messaging on change scores in belief (b = -0.01, SE = 0.05, p = 0.920), worry (b = <-0.01, SE = 0.04, p = 0.928), policy support (b = -0.01,

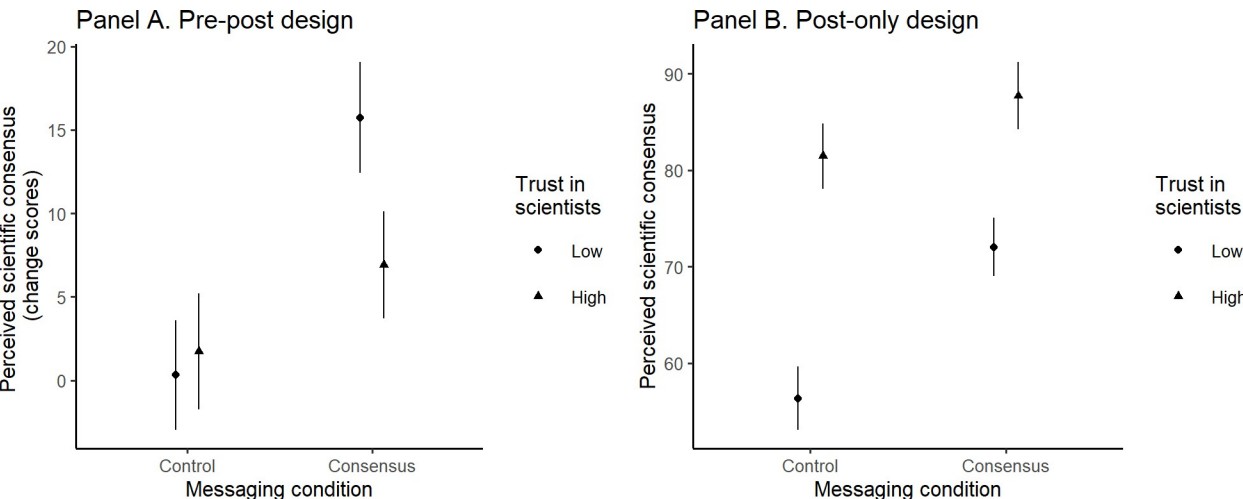

**Fig 7.** The effects of consensus messaging and trust in scientists on perceived consensus in Study 3 in pre-post design (Panel A) and post-only design conditions (Panel B).

SE = 0.03, p = 0.835), vaccination intention with a first dose (b = -0.05, SE = 0.04, p = 0.243), or a booster shot (b = —0.06, SE = 0.06, p = 0.357) were non-significant. Similarly, in the "post-only" design condition the effects of consensus messaging on belief (b = -0.09, SE = 0.09, p = 0.347), worry (b = 0.07, SE = 0.10, p = 0.466), policy support (b = -0.09, SE = 0.11, p = 0.431), vaccination intention with a first dose (b = -0.10, SE = 0.14, p = 0.485), or a booster shot (b = -0.14, SE = 0.13, p = 0.273) were also non-significant. Further details are presented in the S18a and S19a Tables in S1 File.

Again, in all likelihood, consensus messaging is more effective in changing the attitudes and intentions among a subsample of people, i.e., those espousing certain ideologies, trust in scientists, or particular priors. Nevertheless, we did not find any consistent evidence for such moderation. The details are presented in section S3.2.3 of the S1 File and S20a - S25b Tables (S1 File).

Furthermore, as already mentioned, the Gateway Belief Model suggests that the impact of consensus messaging may be indirect. To test this, we ran SEM models separately for the "pre-post" design condition (using change scores) and the "post-only" design condition (using raw scores). Although the model for the "pre-post" design condition had a satisfactory fit (S26 Table in S1 File), the effects of interest were non-significant, replicating the findings from Study 2 (S27 Table in S1 File). For exploratory purposes, we replaced policy support with vaccination intentions, but this analysis did not show promising results either (S28 Table in S1 File). For the "post-only" design condition, the model fit was satisfactory for the model with policy support, or somewhat satisfactory for the model with vaccination intentions (S26 Table in S1 File). These models showed indirect effects of consensus messaging on vaccine belief, worry, and policy support or vaccination intention (S29 and S30 Tables in S1 File). In light of no evidence for sensitization effects, null effects in the analysis with change scores, and the recommendation analyze change scores ("difference-in-difference approach") when testing Gateway Belief Model [62], treating this sole effect as evidence for the effectiveness of consensus messaging does not seem justified.

**Consensus messaging increases psychological reactance but no evidence that reactance is related to changes in vaccine attitudes and intentions.** Moving on to the analysis of psychological reactance, we analyzed all the observations together and modeled the effect of the

design condition. This analysis was not pre-registered but allowed for the verification of whether reactance is higher in the "pre-post" (repeated-measure) design condition than in the "post-only" design condition. The interaction between design and messaging conditions was non-significant (b = 0.11, SE = 0.11, t = 0.97, p = .331); thus, we removed it from the model and re-ran it. Psychological reactance scores were higher in the consensus messaging vs. control condition (b = 0.42, SE = 0.13, t = 7.76, p < .001) but surprisingly, and against sensitization hypothesis, the scores were also slightly higher in the "post-only" vs. "pre-post" design condition (b = 0.13, SE = 0.05, t(1045) = 2.30, p = .022); the details are reported in S31a Table in S1 File.

In addition, we ran analyses with ideology, trust in scientists and priors added to the model, and found that right-wing ideology is positively related to reactance (b = 0.69, SE = 0.21, t(1042) = 3.30, p = .001), whereas high trust in scientists (b = -0.73, SE = 0.14, t(1042) = -5.37, p < .001) and high priors (on scientific consensus; b = -0.67, SE = 0.21, t(501) = -3.16, p = .002) were negatively related to psychological reactance. Notably, the association between these traits and reactance was stronger in the consensus messaging condition in comparison to the control condition (marginally for ideology: b = 0.47, SE = 0.24, t(1042) = 1.96, p = .050; trust in scientists: b = -1.03, SE = 0.19, t(1042) = -5.57, p < .001; priors: b = -1.25, SE = 0.29, t(501) = -4.27, p < .001). The model details, and simple slopes analysis, is shown in section S3.4.1 of the S1 File and S31a Table (S1 File).

Finally, we also conducted exploratory (not pre-registered) analyses of the effects of psychological reactance on change scores in consensus perception, vaccine attitudes and vaccination intentions. These analyses showed no main or interactional effects with the messaging condition, except for the intention to receive a booster shot; surprisingly, the effects of psychological reactance were observed in the control rather than in the consensus message condition. The details are shown in section S3.4.2 of S1 File, S32a Table, and S9 Fig (S1 File).

## General discussion

This research examined the relationship between perceived scientific consensus and attitudes toward the vaccine and vaccination intentions in the context of COVID-19 disease. We aimed to answer seven research questions across three studies, but in fact, fewer questions were able to be addressed with supporting evidence (see Table 1 for summary). Specifically, we discovered that perceived scientific consensus, and vaccine attitudes and intentions were highly associated. Furthermore, single and brief consensus messaging changed perceptions on the safety and efficacy of the vaccines, with the effects lasting at least one week. However, consensus messaging, either in a simple form, or else coupled with a message acknowledging a plurality of opinions on the vaccine, did not yield direct or indirect effects on changes in vaccine attitudes or vaccination intentions.

These null effects of consensus messaging on vaccine attitudes and intentions are surprising given the relatively robust evidence of consensus messaging effectiveness in the pre-pandemic literature; for an overview, see [22]. Furthermore, a recent review of the effectiveness of psychological interventions to mitigate the spread of COVID-19 disease suggests that communicating norms more broadly, and scientific consensus in particular, leads to attitudes and behavioral intentions becoming more aligned with the norms [63]. For example, Bartoš and colleagues [64] demonstrated that when the public is informed about the actual consensus among doctors (i.e. that 90% believe the vaccines are safe), this increases vaccine uptake. This could imply that perceived consensus among practitioners is more instrumental in changing vaccine attitudes than consensus among medical researchers. Previous findings, on the other hand, have indicated that communicating the consensus prevalent among medical scientists

corrected misperceptions about a vaccine-autism link [11, 33], or increased support for COVID-19 prevention policies [12].

Furthermore, the paper by Białek and colleagues [65], which includes a Polish sub-sample, is of special interest here. According to the authors, communicating consensus yields a small positive effect among people with low and medium priors, but the effect is non-significant among people with high priors. Our findings do not support this but we might note that the study by Białek et al. differs in several ways from the current research, with the timing, we believe, being the most important difference between the studies. While their study was run in July, shortly after vaccines became available in May 2021, our experiments were conducted around 5 (Study 2) and 12 months (Study 3) later. This might speak to the different effects of consensus messaging given the context and the novelty of the issue. Findings from Białek et al. [65] (in contrast to ours) could indicate that consensus messaging is more effective in the early stages of a crisis before attitudes have crystalized, at a time when the public is largely confused or uncertain. Alternatively, it might be related to vaccine scarcity at the initial stages of the roll-out. Furthermore, it is worth noting that while the abovementioned research focuses on the effects of expert consensus, whether medical practitioners or scientists, it is likely that perception of social norms might influence vaccine attitudes and intentions. In this context especially relevant are norms within closest social circle and political in-groups [66]. Finally, while we treat trust in scientists as a moderator of the relationship between perceived consensus and vaccine attitudes, it is also likely that trust in scientists changes an issue becomes increasingly politicized [32, 67] or consensus messaging itself might change perception of scientists [68]. The question of what is the dynamics of consensus messaging effectiveness is could be an intriguing avenue for future research.

Still, our study is not alone in its finding that single consensus messaging is ineffective. Chockalingam and colleagues [69] reported, in the context of the COVID-19 pandemic, that consensus messaging about the disease's threat and immunization had null effects on correcting misinformation. According to the authors, the literature on consensus messaging may have overstated the effectiveness of such interventions, and the current findings are in line with this claim.

To examine whether the null effects of consensus messaging are a methodological artifact of repeatedly measuring attitudes, in Study 3, participants were exposed to key DVs before and after consensus messaging, or only after the consensus messaging. In contrast to evidence from Landrum et al. (2021), we did not find evidence for sensitization effects: whether people had seen these questions or not beforehand did not seem to affect how they responded. As a result, it is unlikely that sensitization caused the consensus messaging to have null effect on vaccine attitudes and intentions.

It is also worth noting that we investigated psychological reactance (in Study 3) and discovered that it is indeed higher among people exposed to COVID-19 messaging, particularly among the political Right, those who distrust science, or those with low prior consensus perceptions. However, reactance is unlikely to explain the consensus messaging null effects, given non-significant association between reactance and vaccine attitudes or intentions. We would also like to emphasize that no backfire effects were observed in any of the studies or the conditions. This supports the findings that, while consensus messaging may increase reactance in certain groups, it is unlikely to backfire [70].

To summarize, while the perceived scientific consensus and vaccine attitudes are strongly related, it is unlikely that ad-hoc and single consensus messaging has a sufficiently powerful effect to alter the public's attitudes and intentions. At least when the issue become polarized and attitudes crystalized. Instead, long-term science communications strategies that foster trust and understanding may prove more effective. The current research extends the previous

findings by applying the Gateway Belief Model to investigate vaccine attitudes and intentions at a time when vaccines are widely available, but vaccination rates are rather low. It also contributes to a relatively small number of studies on consensus messaging in the context of health and samples from non-English speaking countries or Western Europe. Finally, it adds to discussions about the effectiveness of communicating norms to change people's attitudes and behaviors, a vital matter in a public health context.

## Supporting information

**S1 File. This file contains all the supporting files.**
(DOCX)

## Acknowledgments

We thank Ilona Iłowiecka-Tańska from Copernicus Science Centre for her support in acquiring funding and conducting this research. We also thank Torun Lindholm Öjmyr and Andreas Olsson for providing feedback on the manuscript.

## Author Contributions

**Conceptualization:** Małgorzata Kossowska.

**Data curation:** Gabriela Czarnek.

**Formal analysis:** Gabriela Czarnek.

**Funding acquisition:** Małgorzata Kossowska.

**Investigation:** Gabriela Czarnek.

**Methodology:** Gabriela Czarnek, Małgorzata Kossowska.

**Visualization:** Gabriela Czarnek.

**Writing – original draft:** Gabriela Czarnek.

**Writing – review & editing:** Gabriela Czarnek, Małgorzata Kossowska.

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
