## [Decision Letter · Decision Letter 0]

23 Oct 2023

PONE-D-23-14532Strong correlational but no causal evidence on the link between the perception of scientific consensus and support for vaccinationPLOS ONE

Dear Dr. Czarnek,

Thank you for submitting your manuscript to PLOS ONE. After careful consideration, we feel that it has merit but does not fully meet PLOS ONE’s publication criteria as it currently stands. Therefore, we invite you to submit a revised version of the manuscript that addresses the points raised during the review process.

1. Kindly reanalyze and present all missing or deleted data to enable full appreciation by readers and future replication by other researchers, with particular reference to the following observations:

2. "Participants who indicated zero consensus". (a) Kindly give reasons for non-inclusion; (b). Rerun the analyses with these zero consensus participants included, and submit a comparative figure or table with those included; Or report these new results in the SI with a footnote referencing it in this section.

3. "Participants whose rating of the manipulation credibility is low". Kindly include these individuals’ responses into the analyses and report the result in the main text.

4.  Removed "speeders (those below the 5 min)". Please add these participants’ results back in to *all* analyses, as well and update the result accordingly.

5. Add a footnote indicating topics or 'unrelated items that were asked on the front end of studies 1 and 2', to give readers better clarity - and then if they wish they can go look up the specific items

6.  Kindly report the results for the 4 out of 6 items that were not significant in the main section, and not the Supplementary material (SI).

7. Kindly include plots or statistical results for the final, key, and best run analysis in study 3.

8. Kindly Run models with and without demographic controls for all results and enter either both results into the relevant results section, or consistently report one set and then consistently just describe the results with language for the second set. Please also have the full models presented in the SI and a footnote to direct the reader

9.  Furthermore, if there was any attrition for any reason, kindly presented at the same location as the sample size and make up. "That is, all studies should have a number that indicates the original sample collected at the start, the final number of participants who completed the study, and if these numbers are not the same, some indication of where these individuals attrited - this last point can be discussed in the SI (referenced by a footnote in the relevant location), but the first two numbers should be placed directly in the sample collection text."

10. Consider including a correlation table that shows the correlations between all the variables mentioned (scientific consensus, worry, belief, policy support, ideology, etc.). 

11. Please consider relabeling the question “Does short messaging change the level of perception of the scientific consensus on vaccine safety and efficacy?” Perhaps this should be labelled as a "manipulation check"?

12. Please address all other comments and observations raised by the peer-reviewers.

We look forward to receiving your revised manuscript.

Kind regards,

Sylvester Chidi Chima, M.D., L.L.M

Academic Editor

PLOS ONE

"The research was supported by the Copernicus Science Centre (Poland). The publication was supported by a grant from the Priority Research Area (Future Society: Behavior in Crisis Lab - Flagship Project) under the Strategic Programme Excellence Initiative at Jagiellonian University (Poland). We thank Ilona Iłowiecka-Tańska for her support in acquiring funding and conducting this research. We also thank Torun Lindholm Öjmyr and Andreas Olsson for providing feedback on the manuscript."

"The research was supported by the Copernicus Science Centre (Poland) https://www.kopernik.org.pl/en. We thank Ilona Iłowiecka-Tańska from Copernicus Science Centre for her support in acquiring funding and data collection.

The publication was supported by a grant from the Priority Research Area (Future Society: Behavior in Crisis Lab - Flagship Project) under the Strategic Programme Excellence Initiative at Jagiellonian University (Poland) https://phils.uj.edu.pl/en_GB/inicjatywa-doskonalosci. This founder had no role in data collection and analysis, decision to publish, or preparation of the manuscript.

We also thank Torun Lindholm Öjmyr and Andreas Olsson for providing feedback on the manuscript."

"The research was supported by the Copernicus Science Centre (Poland) https://www.kopernik.org.pl/en. We thank Ilona Iłowiecka-Tańska from Copernicus Science Centre for her support in acquiring funding and data collection.

The publication was supported by a grant from the Priority Research Area (Future Society: Behavior in Crisis Lab - Flagship Project) under the Strategic Programme Excellence Initiative at Jagiellonian University (Poland) https://phils.uj.edu.pl/en_GB/inicjatywa-doskonalosci. This founder had no role in data collection and analysis, decision to publish, or preparation of the manuscript.

We also thank Torun Lindholm Öjmyr and Andreas Olsson for providing feedback on the manuscript."

**Comments to the Author**

1. Is the manuscript technically sound, and do the data support the conclusions?

Reviewer #1: Yes

Reviewer #2: Yes

Reviewer #3: Partly

2. Has the statistical analysis been performed appropriately and rigorously? 

Reviewer #1: Yes

Reviewer #2: Yes

Reviewer #3: Yes

3. Have the authors made all data underlying the findings in their manuscript fully available?

Reviewer #1: Yes

Reviewer #2: Yes

Reviewer #3: Yes

4. Is the manuscript presented in an intelligible fashion and written in standard English?

Reviewer #1: No

Reviewer #2: Yes

Reviewer #3: Yes

5. Review Comments to the Author

Reviewer #1: This is an excellent manuscript describing several sound studies. My only issue with the study is that it treats trust in scientists as exogenous to COVID-19 attitudes and mitigation behaviors. I would suspect that people became less trusting of scientists as COVID-19 became politicized. However, the authors are consistent with the broader literature in treating it as exogenous. I wouldn't penalize these authors for something although maybe the authors can find work showing the dynamics or changes in trust toward scientists as a result of COVID-19.

In regards to the broader project, more integration of norms would be helpful. The authors focus on consensus at the elite level, but counter views existed creating in-group norms among other elites or peers. Future work should consider this dynamic.

Reviewer #2: The authors of the paper run a correlational study and two survey experiments evaluating the relationship between scientific consensus about vaccines, vaccination attitudes, and intentions. The research is thorough and addresses a question of high practical importance. I think this work would fit well in PLOS ONE.

There are a few changes that I would like to suggest:

Because Study 1 is a correlational study, it would be helpful to include a correlation table that shows the correlations between all the variables mentioned (scientific consensus, worry, belief, policy support, ideology, etc.). It makes it difficult to interpret the results when there is no correlation table.

In Table 1, the second question is: “Does short messaging change the level of perception of the scientific consensus on vaccine safety and efficacy?” However, this seems more of a manipulation check than a research question. I would re-label this as a manipulation check.

In Study 2, 134 participants were dropped for low ratings of manipulation credibility. Is this post-treatment? If so, then differential attrition could explain the results. Please indicate if this is pre or post treatment, and if post-treatment, it would be helpful to run a robustness check including these respondents.

I do not see the pre-registration files in the OSF. Please include links to the pre-registrations in the manuscript.

One reason for the null effects may be that for a salient issue like COVID, a one-sentence message is simply not strong enough to change behaviors. It would be helpful to add this as a limitation.

Reviewer #3: Overall this is a thoughtful test of the role of individual perceptions of the scientific consensus on reported willingness to get vaccinated - in the contexts of Poland. The authors found that, though there is clear evidence for a correlational link between attitudes of scientific consensus and willingness to get vaccinated, there appears to be no causal link, as evidenced by the authors ability to move the former but not the later with a series of experimental studies using consensus messaging. Overall I am positive about this manuscript and will suggest publication once the following changes have been made.

On line 127 the authors indicate that they removed participants who indicated zero consensus. A reason was not given, but I assume it's because the authors believed these individuals were not paying attention or were being insincere in their answer? It could also be that they were engaging in some form of expressive responding. What do the results look like with these individuals added? It seems like this would be the cleanest way of running the analysis - not top down filtering. Though removing these individuals may work against the authors’ position, it still seems like not the cleanest choice.

The authors should:

1- Indicate their reasoning for this decision.

2- Rerun the analyses with these zero consensus participants included

3- And then report these new results in the SI with a footnote referencing it in this section.

Similarly, on line 269 you remove participants whose rating of the manipulation credibility is low, but that’s a violation of randomization and messes with internal validity. That is, these individuals easily could be very low on trust in scientists, trust in the vaccine or both - which could be why they think the message is low. So removing them both violates random assignment but also (and again) works against the authors’ conclusion.

The authors also seem to note this in the intro to study 3, but strangely don't seem to just change the analysis even though (I assume) they have the data?

The authors should:

1- include these individuals’ responses into the analyses and report the result in the main text

Also, removing speeders (those below the 5 min) is a failure of random assignment for the same above reason.

1- Please add these participants’ results back in to *all* analyses, as well and update the results.

The authors indicate several times that other, unrelated items were asked on the front end of studies 1 and 2 and then point to the pre-registration. I think it's fair to say that almost no one will look at this and the authors should instead:

1- add in a footnote the topic of the items that were asked to give the reader some easy understanding - and then if they wish they can go look up the specific items

On line 363 the authors note that 4 out of 6 of the results were not sig, but they do not report them and only report on the two that were significant.

The authors should:

1- Report the results for the 4 that were not sig in the main section, not the SI

I do think that the same in S2 was a bit small - for a test between two means and using the original same size makes sense, but still the results do appear like they would be sig with a larger sample.

It's somewhat strange that the authors did not include plots or statistical results for the final, key, and best run analysis in study 3.

The authors need to:

1- add in these results to make it easier for the reader to understand what they have done

2- they should also include a plot as with all the other key results

Overall, these models are compelling (my critiques above withstanding); however, there are analyses in the SI that seem to use demographic controls in the model - I would request that the authors:

1- Run models with and without demographic controls for all results

2- Enter either both results into the relevant results section, or consistently report one set and then consistently just describe the results with language for the second set

3- have the full models presented in the SI and a footnote to direct the reader

Lastly, if there was any attrition for any reason, I will request these numbers be presented at the same location as the sample size and make up. That is, all studies should have a number that indicates the original sample collected at the start, the final number of participants who completed the study, and if these numbers are not the same, some indication of where these individuals attrited - this last point can be discussed in the SI (referenced by a footnote in the relevant location), but the first two numbers should be placed directly in the sample collection text.

6. PLOS authors have the option to publish the peer review history of their article (what does this mean?). If published, this will include your full peer review and any attached files.

Reviewer #1: No

Reviewer #2: No

Reviewer #3: No

---

## [Author Response · Author response to Decision Letter 0]

10 Nov 2023

Responses has been submitted in a separate file.

---

## [Decision Letter · Decision Letter 1]

6 Dec 2023

Strong correlational but no causal evidence on the link between the perception of scientific consensus and support for vaccination

PONE-D-23-14532R1

Dear Dr. Czarnek,

We’re pleased to inform you that your manuscript has been judged scientifically suitable for publication and will be formally accepted for publication once it meets all outstanding technical requirements.

Kind regards,

Sylvester Chidi Chima, M.D., L.L.M.

Academic Editor

PLOS ONE

Reviewers' comments:

Reviewer's Responses to Questions

**Comments to the Author**

1. If the authors have adequately addressed your comments raised in a previous round of review and you feel that this manuscript is now acceptable for publication, you may indicate that here to bypass the “Comments to the Author” section, enter your conflict of interest statement in the “Confidential to Editor” section, and submit your "Accept" recommendation.

Reviewer #2: All comments have been addressed

Reviewer #3: All comments have been addressed

2. Is the manuscript technically sound, and do the data support the conclusions?

Reviewer #2: Yes

Reviewer #3: Yes

3. Has the statistical analysis been performed appropriately and rigorously? 

Reviewer #2: Yes

Reviewer #3: Yes

4. Have the authors made all data underlying the findings in their manuscript fully available?

Reviewer #2: Yes

Reviewer #3: Yes

5. Is the manuscript presented in an intelligible fashion and written in standard English?

Reviewer #2: Yes

Reviewer #3: Yes

6. Review Comments to the Author

Reviewer #2: The authors have addressed all the comments in my prior referee response. I have no additional comments at this time.

Reviewer #3: This updated version of the manuscript looks in good order and all comments have been responded to/ resolved. I’m happy to endorse this for publication.

7. PLOS authors have the option to publish the peer review history of their article (what does this mean?). If published, this will include your full peer review and any attached files.

Reviewer #2: No

Reviewer #3: No

---

## [Editor Report · Acceptance letter]

20 Dec 2023

PONE-D-23-14532R1 

PLOS ONE

Dear Dr. Czarnek, 

I'm pleased to inform you that your manuscript has been deemed suitable for publication in PLOS ONE. Congratulations! Your manuscript is now being handed over to our production team.

Kind regards, 

on behalf of

Professor Sylvester Chidi Chima 

Academic Editor

PLOS ONE